

# Assessment of the association between periodontal disease and total cancer incidence and mortality: a meta-analysis

Kaili Wang[1], Zheng Zhang[2,3] and Zuomin Wang[4]

[1] Department of Stomatology, Beijing You 'an Hospital, Capital Medical University, Beijing, China
[2] Tianjin Stomatological Hospital, School of Medicine, Nankai University, Tianjin, China
[3] Tianjin Key Laboratory of Oral and Maxillofacial Function Reconstruction, Tianjin, China
[4] Department of Stomatology, Beijing Chao-Yang Hospital, Capital Medical University, Beijing, China

## ABSTRACT

**Background**. Periodontal disease (PD) is a chronic inflammatory disease that leads to alveolar bone resorption and tooth loss. Many studies have reported the association between periodontal disease and various cancers including oral cancer, lung cancer, breast cancer and so on. However, there is still no specialized meta-analysis that assesses the association between periodontal disease and cancer incidence and mortality in-deepth. Thus, we conducted this meta-analysis.

**Methods**. This meta-analysis was registered with PROSPERO: CRD42020183497. We searched five online databases for observational studies about the association between periodontal disease and breast, prostate, lung and bronchial, colorectal, and total cancers by July 2020. Then we evaluated quality of the included studies by the Newcastle-Ottawa scale. Risk ratios (HRs) and their 95% confidence intervals (CIs) were pooled to evaluate the strength of the association between periodontal disease and four cancers, total cancer incidence and mortality. In addition, we analyzed heterogeneity by subgroup analysis and sensitivity analysis. Finally, we inspected publication bias by Begg's and Egger's tests.

**Results**. None of the studies included in this meta-analysis were of poor quality. PD is not only related to breast cancer incidence (HR = 1.26,95%CI [1.11–1.43], $I^2 = 75.8\%$, $P = 0.000$), but also connected with total cancer mortality (HR = 1.40,95%CI [1.24–1.58], $I^2 = 0.0\%$, $P = 0.718$). Subgroup analyses showed that study population, study design, dental status, follow-up period, adjustment for smoking partially explained the heterogeneity between studies. The results of Begg's test and Egger's test were consistent and indicated that there is no publication bias in this study.

**Conclusion**. In conclusion, this meta-analysis revealed a positive relationship between periodontal disease and breast cancer incidence and total cancer mortality. Further well-designed studies with specific inclusion and exclusion criteria are required to strengthen the conclusion of this meta-analysis. However, longer follow-up period, multi-center trials and even multinational studies are required to corroborate the results.

Corresponding authors
Zheng Zhang,
zhangzheng@nankai.edu.cn
Zuomin Wang, wzuomin@sina.cn

## INTRODUCTION

Periodontal disease is a chronic inflammatory disease that eventually leads to tooth loss. The initial factor associated with periodontal disease is dental plaque, which attacks the first immune defense line of periodontal tissue, leading to inflammation and disease. Periodontitis has a range of clinical manifestations, including gingiva bleeding, periodontal pocket, alveolar bone resorption, and tooth loss. Periodontitis is the major cause of tooth loss in adults. According to a National Health and Nutrition Examination survey, approximately 42% of American adults aged 30 years or older had periodontitis, with 7.8% having severe periodontitis (*Eke et al., 2018*). In Turkey, the proportion of individuals with more than three mm loss of attachment (LOA) ranged from 43% to 91% (*Ilhan et al., 2017*). In the Fourth National Oral Health Survey, the prevalence of probe bleeding in the 35 to 44-year-old population was 87.4% (*Sun et al., 2018*). There is a verified association between periodontal disease and a variety of diseases, such as rheumatoid arthritis (*Molon et al., 2019*), diabetes (*Genco & Borgnakke, 2020*), cardiovascular disease (*Sanz et al., 2020*), chronic obstructive pulmonary disease (*Takeuchi et al., 2019*), depression (*Nascimento et al., 2019*), and Alzheimer's disease (*Liccardo et al., 2020*).

Periodontitis is not only related to the abovementioned diseases but also to cancers. In recent years, an increasing number of studies have revealed the relationship between periodontal disease and various cancers, particularly head and neck (*Eliot et al., 2013*), esophageal (*Malinowski et al., 2019*), and pancreatic (*Fan et al., 2018*) cancers. Subsequently, researchers conducted meta-analyses of previous studies in order to reach a more accurate and objective conclusion. For example, *Yao et al. (2014)* reviewed five reliable studies, obtained data through the random effect model, and finally concluded that patients with periodontitis were prone to oral cancer. Six years later, *Gopinath et al. (2020)* analyzed nine studies and confirmed a significant correlation between periodontitis and head and neck cancer.

Cancer is not only difficult to overcome but also a heavy economic burden. In China (*Cao et al., 2020*), lung cancer remains the most common cancer and accounts for nearly 30% of cancer deaths. In 2015, the five cancers with the highest incidence included lung, stomach, colorectum, liver, and breast, accounting for nearly 60% of all diagnosed cancers. Every year, the American Cancer Society estimates the number of new cancer cases and cancer-related deaths in the population. According to recent data collected in 2019, prostate and lung & bronchial cancers were the two most common cancers in men, while, breast, lung, uterine corpus and colorectal cancers were the four most common cancers in women. The largest numbers of deaths result from lung, prostate, and colorectal cancers in men and lung, breast, and colorectal cancers in women (*Siegel, Miller & Jemal, 2019*).

In this context, we aimed to assess the correlations, in terms of incidence and mortality, between periodontal disease and cancers, including breast, prostate, lung and bronchial, colorectal, and total cancers, as well as the strength of these correlations. Furthermore, we aimed to preliminarily explore the factors that influence these correlations.
## MATERIAL AND METHODS

### Methodology

This meta-analysis was reported according to the Preferred Reporting Items for Systematic Review and Meta-analysis (PRISMA) guidelines. The protocol was registered in PROSPERO (http://www.crd.york.ac.uk/PROSPERO) before the research was begun (CRD 42020183497).

### Criteria for the inclusion of studies

The following criteria were applied: (1) studies published in English or Chinese; (2) studies on human subjects; (3) observational studies (including cohort and case-control studies)with a prospective or retrospective design; (4) studies with periodontal disease as the exposure and, the incidence or mortality associated with breast or prostate cancer, lung& bronchial or colorectal cancer, or total cancers as the outcome; and (5) studies providing relative risks (RRs), hazard ratios (HRs), or odds ratios (ORs) and corresponding 95% confidence intervals (CIs).

### Literature search and selection

Databases: PubMed, Embase, Web of Science, China National Knowledge Infrastructure [CNKI] and Wanfang.

Search terms: (<Neoplasm= OR <Carcinoma= OR <cancer= OR <tumor= OR <tumour=) and (<periodontal diseases= OR <gum disease= OR <periodontitis= OR <gingivitis= OR <periodontal= OR <periodontium= OR <periodontal attachment loss= OR <periodontal pocket= OR <alveolar bone loss= OR <tooth mobility=).

Time: by to July 2020.

First, two authors (Kaili Wang and Zheng Zhang) independently retrieved studies from the databases using specific search terms. Subsequently, a hand search was performed by reading abstracts and key words, and studies that were eligible for the inclusion criteria were selected. Finally, a consensus regarding the final studies to be included was reached *via* a discussion or by consulting a third author (Zuomin Wang).

### Data extraction

Kaili Wang sorted out the information for the subsequent classification of the included studies: first author, publication date, country/region, study design type, numbers of samples, age range of the study population, diagnostic criteria for periodontal disease and cancer, cancer type and outcomes, follow-up period, whether adjustments were performed for smoking, adjusted variables, and multiple adjusted HRs and corresponding 95% CIs. Next, Zheng Zhang checked the information and supplemented it if necessary.

### Quality assessment

We evaluated the quality of the included studies using the Newcastle-Ottawa scale. This evaluation system includes three aspects: selection, comparability and exposure. In the selection part, we evaluated the following four aspects: whether the exposed cohort could basically represent the average of the community population, whether the non-exposed cohort was from the same source as the exposure cohort, whether the selection of the

exposed cohort was reliable based on records, and whether no outcome events occurred in the study subjects before the study began. In the comparability part, we assessed whether the exposed and non-exposed cohorts were matched in terms of age and other confounding factors. In the exposure part, we assessed the following three aspects: whether there were reliable data or records, such as examination or medical records, for the evaluation of outcome events; whether the follow-up period was equal to or longer than 5 years; and the completeness of follow-up through factors, such as the loss to follow-up rate.

Based on the scores, the studies were defined as follows: <4 points, low quality; 4–6 points, moderate quality; and 7–9 points, high quality.

### Classification of studies

First, the included studies were classified according to the study scope: breast, prostate, lung & bronchial, colon & Rectum, and total cancers. Next, according to the specific content and purpose of the studies, we divided them based on two aspects: cancer occurrence and cancer mortality. Of course, som studies involved both incidence and mortality.

### Statistical analysis

We used the HR with the 95% CI to assess the association between periodontal disease and the risk of cancer incidence and mortality. The data extracted from the studies were analyzed using STATA software (version 12.0; Stata Corp, College Station, TX, USA). Heterogeneity was tested through the $Q$ test (statistical significance was set at $P < 0.1$) and $I^2$ test ($I^2 \geq 50\%$ indicated significant heterogeneity) and was represented using a forest plot.

Subgroup analyses were conducted to explore the factors influencing heterogeneity. Subgroup analyses were stratified by study population, study design, dental status, follow-up period, and whether or not smoking was adjusted for. A sensitivity analysis was performed to test the stability of the results.

Begg's and Egger's tests were used for assessing publication bias. Begg's test, also known as the rank correlation test, tests the correlation between effects and sample size. Egger's test, also known as the linear regression method, establishes a regression equation based on the standard normal deviate and precision of each included study. The larger the intercept of the regression line, the higher the bias. A $P$-value of $> 0.05$ was considered to indicate no publication bias..

## RESULTS

### Search and selection results

The process of study selection is shown in Fig. 1. First, 3,324 records were collected from PubMed, 5,681 from Embase, 5,435 from Web of Science, 1,095 from Wanfang, and 1,576 from CNKI. Subsequently, 9,860 publications remained after the duplicates were removed, of which 72 were deemed eligible based on their abstracts and key words. Finally, 27 publications were included in the meta-analysis after 45 studies were excluded for several reasons. Duplicates were removed for the following reasons: review/commentary/letter ($n = 13$), repeat publication ($n = 7$), no available data ($n = 22$), and no full text ($n = 3$).

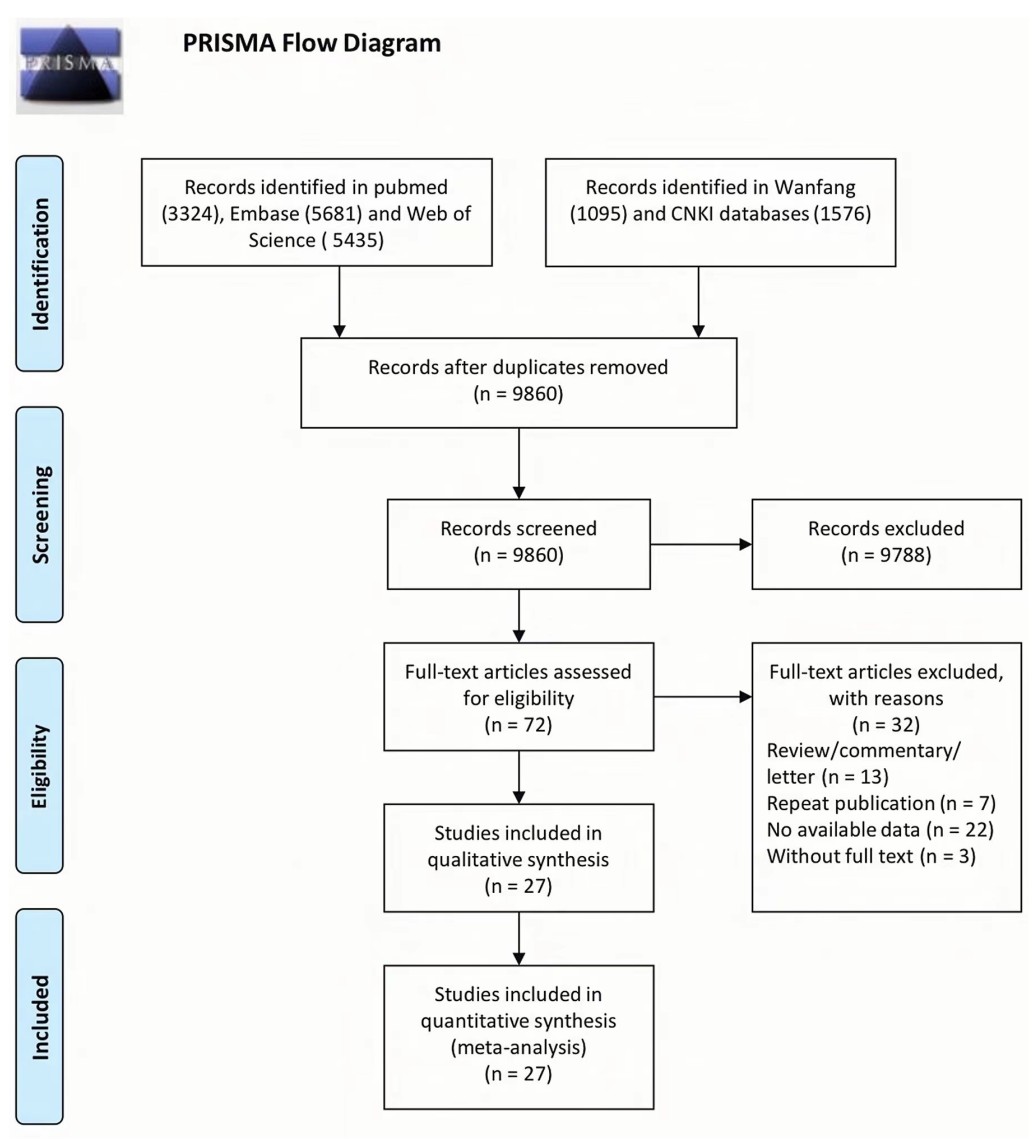

**Figure 1 Flow chart from identification of eligible studies to final inclusion.**

## Characteristics of the included studies

Table 1 describes the studies included in the review. The articles were published between 2003 and 2020. Twelve studies were conducted in North America and six in China. Six studies were retrospective, and the others were prospective. Periodontal disease was diagnosed through a dental examination in 18 studies (*Hujoel et al., 2003*; *Ahn, Segers & Hayes, 2012*; *Wen et al., 2014*; *Chrysanthakopoulos, 2016*; *Chung et al., 2016*; *Mai et al., 2016*; *Dizdar et al., 2017*; *Han, 2017*; *Lee et al., 2017*; *Sfreddo et al., 2017*; *Chou et al., 2018*; *Heikkilä et al., 2018*; *Hu et al., 2018*; *Michaud et al., 2018*; *Güven et al., 2019*; *Lu et al., 2019*; *Tai et al., 2019*; *Huang et al., 2020*), among which the International Classification of Diseases, Ninth Revision, Clinical Modification (ICD-9-CM) was used in five studies

(*Wen et al., 2014*; *Chung et al., 2016*; *Chou et al., 2018*; *Hu et al., 2018*; *Tai et al., 2019*), and the definition criteria of periodontal pocket and attachment loss were used in 13 studies (*Hujoel et al., 2003*; *Ahn, Segers & Hayes, 2012*; *Chrysanthakopoulos, 2016*; *Mai et al., 2016*; *Dizdar et al., 2017*; *Han, 2017*; *Lee et al., 2017*; *Sfreddo et al., 2017*; *Heikkilä et al., 2018*; *Michaud et al., 2018*; *Güven et al., 2019*; *Lu et al., 2019*; *Huang et al., 2020*). The diagnosis was self-reported by patients based on questionnaires and tooth loss history in the remaining nine studies (*Michaud et al., 2008*; *Arora et al., 2010*; *Mai et al., 2014*; *Freudenheim et al., 2016*; *Michaud et al., 2016*; *Momen-Heravi et al., 2017*; *Nwizu et al., 2017*; *Lee et al., 2018*; *Yoon et al., 2019*), among which dentists validated the diagnosis in three studies (*Michaud et al., 2016*; *Momen-Heravi et al., 2017*; *Lee et al., 2018*). Six studies analyzed periodontal disease and cancer mortality (*Hujoel et al., 2003*; *Ahn, Segers & Hayes, 2012*; *Mai et al., 2014*; *Heikkilä et al., 2018*; *Lu et al., 2019*; *Huang et al., 2020*), and 22 studies analyzed periodontal disease and cancer incidence (*Michaud et al., 2008*; *Arora et al., 2010*; *Wen et al., 2014*; *Chrysanthakopoulos, 2016*; *Chung et al., 2016*; *Freudenheim et al., 2016*; *Mai et al., 2016*; *Michaud et al., 2016*; *Dizdar et al., 2017*; *Han, 2017*; *Lee et al., 2017*; *Momen-Heravi et al., 2017*; *Nwizu et al., 2017*; *Sfreddo et al., 2017*; *Chou et al., 2018*; *Hu et al., 2018*; *Lee et al., 2018*; *Michaud et al., 2018*; *Güven et al., 2019*; *Lu et al., 2019*; *Tai et al., 2019*; *Yoon et al., 2019*). Cancer diagnosis was based on ICD-9/10-CM in 15 studies (*Hujoel et al., 2003*; *Arora et al., 2010*; *Ahn, Segers & Hayes, 2012*; *Mai et al., 2014*; *Wen et al., 2014*; *Chung et al., 2016*; *Lee et al., 2017*; *Momen-Heravi et al., 2017*; *Nwizu et al., 2017*; *Sfreddo et al., 2017*; *Chou et al., 2018*; *Heikkilä et al., 2018*; *Hu et al., 2018*; *Tai et al., 2019*; *Yoon et al., 2019*), medical records and histological examination in 11 studies (*Michaud et al., 2008*; *Chrysanthakopoulos, 2016*; *Freudenheim et al., 2016*; *Mai et al., 2016*; *Michaud et al., 2016*; *Dizdar et al., 2017*; *Lee et al., 2018*; *Michaud et al., 2018*; *Güven et al., 2019*; *Lu et al., 2019*; *Huang et al., 2020*), and patient self-report in 1 study (*Han, 2017*). All the participants were aged above 18 years, and the follow-up period was more than 10 years in 15 studies (*Hujoel et al., 2003*; *Michaud et al., 2008*; *Arora et al., 2010*; *Ahn, Segers & Hayes, 2012*; *Wen et al., 2014*; *Mai et al., 2016*; *Michaud et al., 2016*; *Dizdar et al., 2017*; *Lee et al., 2017*; *Momen-Heravi et al., 2017*; *Chou et al., 2018*; *Heikkilä et al., 2018*; *Michaud et al., 2018*; *Tai et al., 2019*; *Huang et al., 2020*) and less than 10 years in 12 studies (*Mai et al., 2014*; *Chrysanthakopoulos, 2016*; *Chung et al., 2016*; *Freudenheim et al., 2016*; *Han, 2017*; *Nwizu et al., 2017*; *Sfreddo et al., 2017*; *Hu et al., 2018*; *Lee et al., 2018*; *Güven et al., 2019*; *Lu et al., 2019*; *Yoon et al., 2019*). Most studies collected information on whether the participants smoked, including how often and how much they smoked and the pack-years. Regarding smoking, eight studies (*Wen et al., 2014*; *Chung et al., 2016*; *Dizdar et al., 2017*; *Chou et al., 2018*; *Heikkilä et al., 2018*; *Hu et al., 2018*; *Güven et al., 2019*; *Tai et al., 2019*) did not perform adjustments for smoking; however, the remaining studies included smokers and took smoking into account when calculating the HRs.

## Quality assessment

Table S1 shows the quality assessment of the included studies. Twenty-four studies scored seven or more stars, indicating that 88.9% of the studies were of good quality.

Wang et al. (2022), *PeerJ*, DOI 10.7717/peerj.14320

**Table 1** Characteristics of eligible studies included in this meta-analysis.

| Author (Year) | Location | Study design | Participants number | Age (years) | Dental status | Outcomes | Cancer type | Assessment method | Follow-up (year) | Adjusted variables |
|---|---|---|---|---|---|---|---|---|---|---|
| Hujoel et al. 2003 | USA | Prospective study | 11,328 | 25–74 | Dental examination | Cancer mortality | breast, prostate, lung & bronchus, colon & rectum, total cancer | ICD-9 | 18 | age, gender, education, socioeconomic level, race, smoking, alcohol consumption, vitamin consumption, et al. |
| Michaud et al. 2008 | USA | Prospective study | 48,375 | 40–75 | Self-reported | Cancer incidence | prostate, lung & bronchus, colon & rectum, total cancer | Medical records or histological examination | 17.7 | age, race, smoking, alcohol consumption, vitamin consumption, body mass index, diabetes, physical activity, et al. |
| Arora et al. 2010 | Swedish | Prospective study | 15,333 | 38–77 | Self-reported | Cancer incidence | breast, prostate, colon & rectum, total cancer | ICD | 27 | age, gender, education, smoking, alcohol consumption, body mass index, diabetes, et al. |
| Ahn et al. 2012 | USA | Prospective study | 12,605 | ≥17 | Dental examination | Cancer mortality | colon & rectum | ICD-10 | 12 | age, gender, education, race, smoking, body mass index, et al. |
| Mai et al. 2014 | USA | Prospective study | 77,485 | 50–79 | Self-reported | Cancer mortality | lung & bronchus | ICD-O-2 | 6.8 | age, education, race, smoking, alcohol consumption, body mass index, physical activity, et al. |
| Wen et al. 2014 | China | Prospective study | 1 million | >20 | Dental examination | Cancer incidence | breast, lung & bronchus, total cancer | ICD-9 | 14 | age, gender, et al. |
| Chrysanthakopoulos et al. 2016 | Greek | Retrospective study | 200 | ≥48 | Dental examination | Cancer incidence | lung & bronchus | Medical records or histological examination | 2 | age, gender, education, socioeconomic level, smoking, et al. |
| Chung et al. 2016 | China | Prospective study | 40,140 | ≥40 | Dental examination | Cancer incidence | breast, total cancer | ICD-9 | 5 | socioeconomic level, diabetes, et al. |
| Freudenheim et al. 2016 | USA | Prospective study | 73,737 | 50–79 | Self-reported | Cancer incidence | breast | Medical records or histological examination | 6.7 | age, education, race, smoking, alcohol consumption, body mass index, physical activity, et al. |
| Mai et al. 2016 | China | Prospective study | 1,337 | 53–85 | Dental examination | Cancer incidence | breast, lung & bronchus, colon & rectum, total cancer | Medical records or histological examination | 12.2 | age, smoking, et al. |
| Michaud et al. 2016 | USA | Prospective study | 19,933 | 40–75 | Self-reported | Cancer incidence | prostate, lung & bronchus, colon & rectum, total cancer | Medical records or histological examination | 26 | age, race, smoking, alcohol consumption, body mass index, diabetes, physical activity, et al. |
| Dizdar et al. 2017 | Turkey | Prospective study | 280 | ≥35 | Dental examination | Cancer incidence | breast, prostate, lung & bronchus, total cancer | Medical records or histological examination | 12 | age, gender, et al. |
| Han et al. 2017 | Korea | Retrospective study | 22,948 | ≥19 | Dental examination | Cancer incidence | breast, lung & bronchus, colon & rectum, total cancer | self-reported | 2 | education, socioeconomic level, smoking, alcohol consumption, et al. |
| Lee et al. 2017 | Korea | Prospective study | 934 | ≥40 | Dental examination | Cancer incidence | prostate, colon & rectum | ICD-10 | 12 | age, gender, socioeconomic level, smoking, alcohol consumption, diabetes, physical activity, et al. |

**Table 1** (*continued*)

| Author (Year) | Location | Study design | Participants number | Age (years) | Dental status | Outcomes | Cancer type | Assessment method | Follow-up (year) | Adjusted variables |
|---|---|---|---|---|---|---|---|---|---|---|
| Momen-Heravi et al. 2017 | USA | Prospective study | 121,700 | 30–55 | Self-reported | Cancer incidence | colon & rectum | ICD-9 | 19 | age, race, smoking, alcohol consumption, vitamin consumption, body mass index, diabetes, et al. |
| Nwizu1 et al. 2017 | USA | Prospective study | 65,869 | 54–86 | Self-reported | Cancer incidence | breast, lung & bronchus, colon & rectum, total cancer | ICD or Medical records | 8.32 | age, smoking, body mass index, et al. |
| Sfreddo et al. 2017 | Brazil | Retrospective study | 201 | ≥18 | Dental examination | Cancer incidence | breast | ICD-10 or Medical records | 2.2 | age, smoking, body mass index, et al. |
| Chou et al. 2018 | China | Retrospective study | 67,672 | ≥18 | Dental examination | Cancer incidence | colon & rectum | ICD-9 | 10 | age, gender, education, socioeconomic level, diabetes, et al. |
| Heikkilä et al. 2018 | Finland | Prospective study | 68,273 | ≥29 | Dental examination | Cancer mortality | breast, prostate, lung & bronchus, total cancer | ICD-10 | 10.1 | age, gender, socioeconomic level, diabetes, et al. |
| Hu et al. 2018 | China | Prospective study | 212,974 | ≥18 | Dental examination | Cancer incidence | colon & rectum | ICD-9 | 3 | age, gender, et al. |
| Lee et al. 2018 | Korea | Retrospective study | 42,871 | 30–45 | Self-reported | Cancer incidence | colon & rectum | Medical records or histological examination | 3 | age,sex,BMI, DM, HbA1c, HTN, LDL ,waist,alcohol intake,smoking status and pack-years. |
| Michaud et al. 2018 | USA | Prospective study | 7,466 | 44–66 | Dental examination | Cancer incidence | breast, prostate, lung & bronchus, colon & rectum, total cancer | Medical records or histological examination | 14.7 | age, field center, education level, smoking status, smoking duration, drinking status, body mass index, and diabetes status |
| Güven et al. 2019 | Turkey | Prospective study | 5,199 | ≥50 | Dental examination | Cancer incidence | lung & bronchus, colon & rectum, total cancer | Medical records or histological examination | 7.2 | age, gender, et al. |
| Lu et al. 2019 | USA | Prospective study | 15,792 | 44–66 | Dental examination | Cancer incidence and mortality | Lung & bronchus, colon & rectum, total cancer | Medical records or histological examination | 4 | age, education, race, smoking, alcohol consumption, diabetes, et al. |
| Tai et al. 2019 | China | Prospective study | 714,246 | No metion | Dental examination | Cancer incidence | lung & bronchus | ICD-9 | ≥10 | no |
| Yoon et al. 2019 | USA | Retrospective study | 84,797 | 40–79 | Self-reported | Cancer incidence | lung & bronchus | ICD-10 | 8 | education, socioeconomic level, smoking, alcohol consumption, body mass index, et al. |
| Huang et al. 2020 | USA | Prospective study | 6,034 | ≥40 | Dental examination | Cancer mortality | total cancer | Medical records or histological examination | 21.3 | age, gender, education, socioeconomic level, race, smoking, body mass index, diabetes, et al. |

**Notes.**

ICD: International Classification of Diseases.

### Association between periodontitis and cancer incidence

The relationship between periodontal disease and cancer incidence is shown in Fig. 2. The pooled estimate based on 11 studies revealed a modest association between periodontal disease and breast cancer incidence (HR = 1.26, 95% CI [1.11–1.43]), with greater heterogeneity ($I^2 = 75.8\%$, $P_Q = 0.000$). Similarly, we found a modest association between periodontal disease and prostate (HR = 1.26, 95% CI [1.03–1.54], $I^2 = 72.9\%$, $P_Q = 0.001$), lung & bronchus (HR = 1.30, 95% CI [1.13–1.50], $I^2 = 63.7\%$, $P_Q = 0.000$), colon & rectum (HR = 1.18, 95% CI [1.03–1.36], $I^2 = 79.0\%$, $P_Q = 0.000$), and total (HR = 1.14, 95% CI [1.08–1.20], $I^2 = 67.4\%$, $P_Q = 0.000$)cancer incidence.

Next, we explored the source of heterogeneity through subgroup analyses with $I^2 > 50\%$ and $P < 0.1$.

### Sensitivity analyses of the association between periodontitis and cancer incidence

As shown in Fig. 3, the pooled HRs were the same, and the 95%CI overlapped with the previous effect size. This means that our results are very reliable.

### Subgroup analyses of the association between periodontitis and cancer incidence

The subgroup analysis results are presented in Table 2.

The following were the observations regarding the 11 studies that were conducted on periodontal disease and breast cancer incidence. Four studies were from North America and showed an HR of 1.12 ($I^2 = 0\%$, 95% CI [1.05–1.19], $P = 0.001$). Two studies were retrospective and showed an HR of 1.52 ($I^2 = 0\%$, 95% CI [1.17–1.98], $P = 0.002$). The pooled HR in four studies in which the periodontal disease diagnosis was self-reported was 1.15 ($I^2 = 0\%$, 95% CI [1.09–1.22], $P = 0.001$). Six studies with follow-up periods of more than 10 years showed an HR of 1.11( $I^2 = 14.4\%$, 95% CI [1.03–1.19], $P = 0.005$). The pooled HR in seven studies in which smoking was adjusted for was 1.13 ($I^2 = 7.7\%$, 95% CI [1.07–1.21], $P = 0.001$). The CIs were all on the right side. The results of the subgroup analyses were consistent with those of the full dataset analysis. We can state with certainty that periodontal disease increases the risk of breast cancer incidence.

The following were the observations in the seven studies on periodontal disease and prostate cancer incidence. Three studies were from North America and showed an HR of 1.03 ($I^2 = 36.9\%$, 95% CI [0.89–1.18], $P = 0.728$). Six studies with follow-up periods of more than 10 years showed an HR of 1.16 ($I^2 = 60.8\%$, 95% CI [0.97–1.38], $P = 0.100$).We found that periodontal disease increased the risk of prostate cancer incidence under certain conditions in the European people, dental examination diagnosis of periodontal disease, follow-up period of less than 10 years. However, the association was not affected by smoking.

Among the 14 studies on periodontal disease and lung and bronchial cancer incidence, eight studies were from North America and showed an HR of 1.34 ($I^2 = 46.9$, 95% CI [1.24–1.46], $P = 0.001$), and eight studies in which periodontal disease was diagnosed through dental examination showed an HR of 1.33 ($I^2 = 75.7$, 95% CI [0.94–1.88],

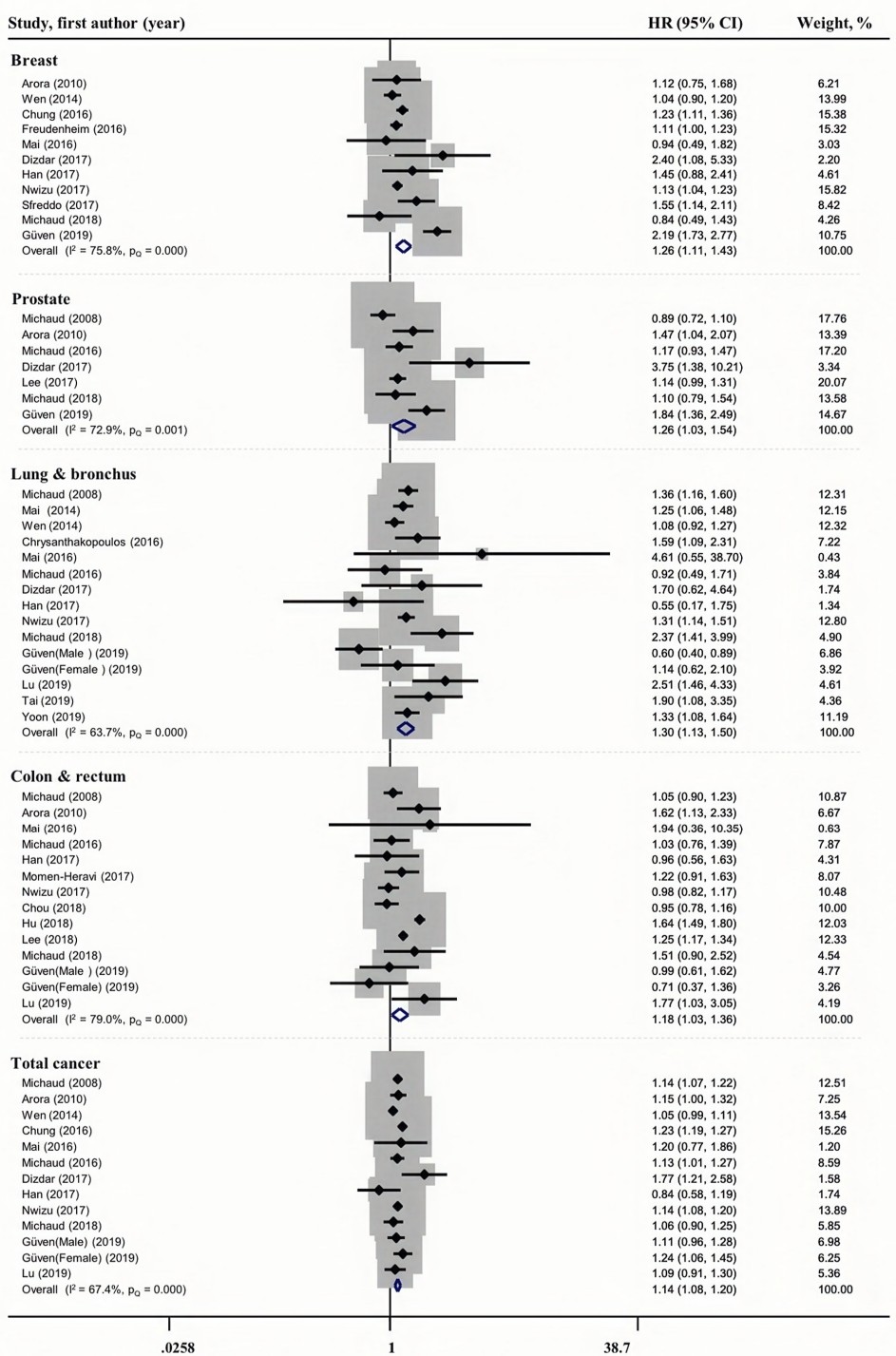

**Figure 2  Forest plots of the association between periodontitis and cancer incidence.**

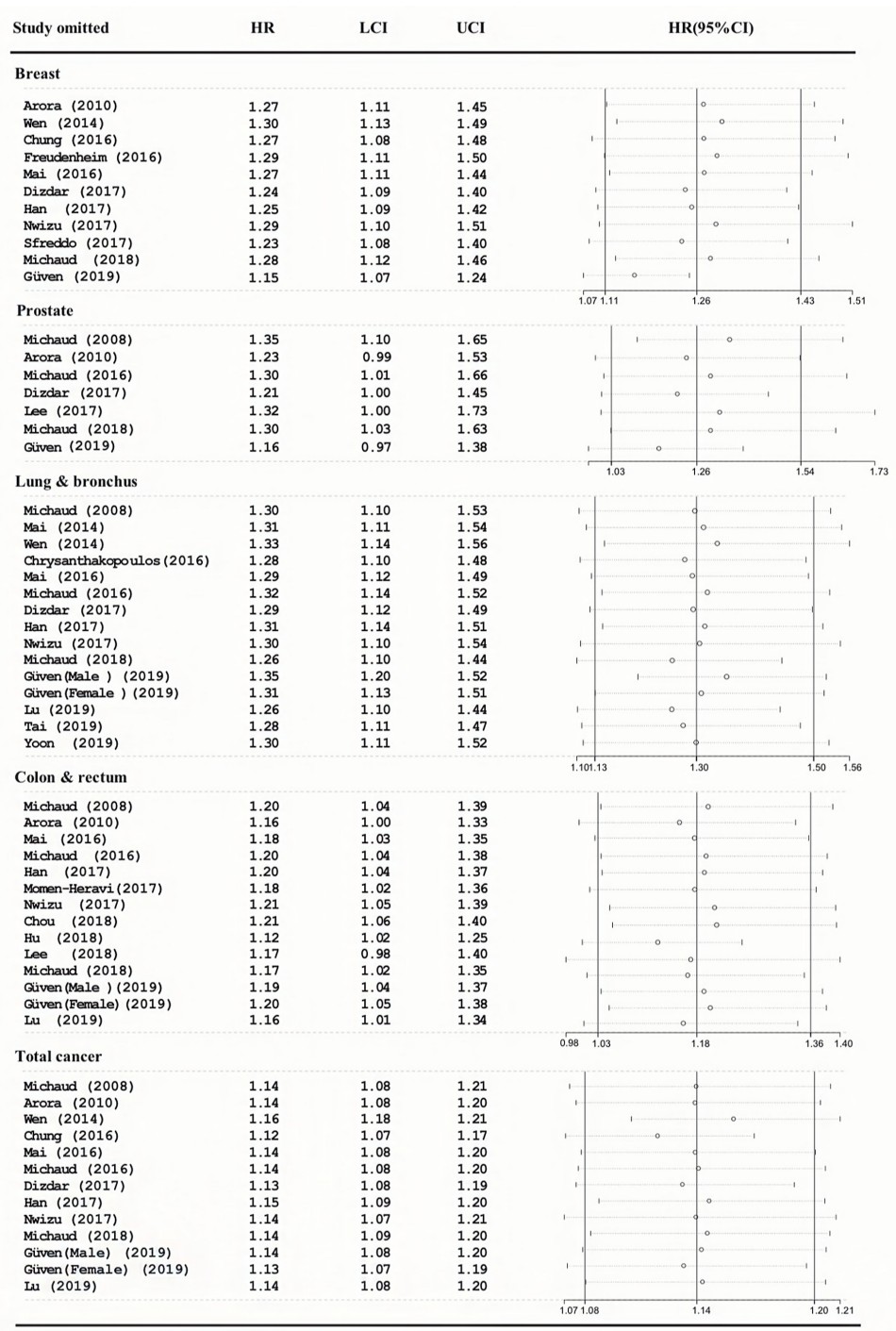

| Study omitted | HR | LCI | UCI | HR(95%CI) |
|---|---|---|---|---|
| **Breast** | | | | |
| Arora (2010) | 1.27 | 1.11 | 1.45 | |
| Wen (2014) | 1.30 | 1.13 | 1.49 | |
| Chung (2016) | 1.27 | 1.08 | 1.48 | |
| Freudenheim (2016) | 1.29 | 1.11 | 1.50 | |
| Mai (2016) | 1.27 | 1.11 | 1.44 | |
| Dizdar (2017) | 1.24 | 1.09 | 1.40 | |
| Han (2017) | 1.25 | 1.09 | 1.42 | |
| Nwizu (2017) | 1.29 | 1.10 | 1.51 | |
| Sfreddo (2017) | 1.23 | 1.08 | 1.40 | |
| Michaud (2018) | 1.28 | 1.12 | 1.46 | |
| Güven (2019) | 1.15 | 1.07 | 1.24 | |
| **Prostate** | | | | |
| Michaud (2008) | 1.35 | 1.10 | 1.65 | |
| Arora (2010) | 1.23 | 0.99 | 1.53 | |
| Michaud (2016) | 1.30 | 1.01 | 1.66 | |
| Dizdar (2017) | 1.21 | 1.00 | 1.45 | |
| Lee (2017) | 1.32 | 1.00 | 1.73 | |
| Michaud (2018) | 1.30 | 1.03 | 1.63 | |
| Güven (2019) | 1.16 | 0.97 | 1.38 | |
| **Lung & bronchus** | | | | |
| Michaud (2008) | 1.30 | 1.10 | 1.53 | |
| Mai (2014) | 1.31 | 1.11 | 1.54 | |
| Wen (2014) | 1.33 | 1.14 | 1.56 | |
| Chrysanthakopoulos (2016) | 1.28 | 1.10 | 1.48 | |
| Mai (2016) | 1.29 | 1.12 | 1.49 | |
| Michaud (2016) | 1.32 | 1.14 | 1.52 | |
| Dizdar (2017) | 1.29 | 1.12 | 1.49 | |
| Han (2017) | 1.31 | 1.14 | 1.51 | |
| Nwizu (2017) | 1.30 | 1.10 | 1.54 | |
| Michaud (2018) | 1.26 | 1.10 | 1.44 | |
| Güven (Male ) (2019) | 1.35 | 1.20 | 1.52 | |
| Güven (Female ) (2019) | 1.31 | 1.13 | 1.51 | |
| Lu (2019) | 1.26 | 1.10 | 1.44 | |
| Tai (2019) | 1.28 | 1.11 | 1.47 | |
| Yoon (2019) | 1.30 | 1.11 | 1.52 | |
| **Colon & rectum** | | | | |
| Michaud (2008) | 1.20 | 1.04 | 1.39 | |
| Arora (2010) | 1.16 | 1.00 | 1.33 | |
| Mai (2016) | 1.18 | 1.03 | 1.35 | |
| Michaud (2016) | 1.20 | 1.04 | 1.38 | |
| Han (2017) | 1.20 | 1.04 | 1.37 | |
| Momen-Heravi (2017) | 1.18 | 1.02 | 1.36 | |
| Nwizu (2017) | 1.21 | 1.05 | 1.39 | |
| Chou (2018) | 1.21 | 1.06 | 1.40 | |
| Hu (2018) | 1.12 | 1.02 | 1.25 | |
| Lee (2018) | 1.17 | 0.98 | 1.40 | |
| Michaud (2018) | 1.17 | 1.02 | 1.35 | |
| Güven (Male ) (2019) | 1.19 | 1.04 | 1.37 | |
| Güven (Female) (2019) | 1.20 | 1.05 | 1.38 | |
| Lu (2019) | 1.16 | 1.01 | 1.34 | |
| **Total cancer** | | | | |
| Michaud (2008) | 1.14 | 1.08 | 1.21 | |
| Arora (2010) | 1.14 | 1.08 | 1.20 | |
| Wen (2014) | 1.16 | 1.18 | 1.21 | |
| Chung (2016) | 1.12 | 1.07 | 1.17 | |
| Mai (2016) | 1.14 | 1.08 | 1.20 | |
| Michaud (2016) | 1.14 | 1.08 | 1.20 | |
| Dizdar (2017) | 1.13 | 1.08 | 1.19 | |
| Han (2017) | 1.15 | 1.09 | 1.20 | |
| Nwizu (2017) | 1.14 | 1.07 | 1.21 | |
| Michaud (2018) | 1.14 | 1.09 | 1.20 | |
| Güven (Male) (2019) | 1.14 | 1.08 | 1.20 | |
| Güven (Female) (2019) | 1.13 | 1.07 | 1.19 | |
| Lu (2019) | 1.14 | 1.08 | 1.20 | |

**Figure 3  Sensitivity analyses of the association between periodontitis and cancer incidence.**

**Table 2 Subgroup analyses of the association between periodontitis and cancer incidence.**

| Subgroups | No. of studies | Meta-analysis | | | Heterogeneity | |
|---|---|---|---|---|---|---|
| | | HR | 95%CI | *P*-value | I²(%) | *P*-value |
| **Breast Cancer** | | | | | | |
| **Study population** | | | | | | |
| North America | 4 | 1.12 | 1.05–1.19 | 0.001 | 0.0 | 0.694 |
| Asia | 3 | 1.16 | 1.00–1.35 | 0.044 | 53.1 | 0.119 |
| Europe | 3 | 1.76 | 1.06–2.91 | 0.029 | 75.8 | 0.016 |
| South America | 1 | 1.55 | 1.14–2.11 | 0.006 | – | – |
| **Study design** | | | | | | |
| Prospective study | 9 | 1.22 | 1.07–1.40 | 0.004 | 78.7 | 0.000 |
| Retrospective study | 2 | 1.52 | 1.17–1.98 | 0.002 | 0.0 | 0.832 |
| **Dental status** | | | | | | |
| Self-reported | 4 | 1.15 | 1.09–1.22 | <0.001 | 0.0 | 0.502 |
| Dental examination | 7 | 1.38 | 1.00–1.90 | 0.051 | 83.1 | 0.000 |
| **Follow-up (years)** | | | | | | |
| ≥10 | 6 | 1.11 | 1.03–1.19 | 0.005 | 14.4 | 0.322 |
| <10 | 5 | 1.43 | 1.15–1.78 | 0.001 | 86.3 | 0.000 |
| **Adjust for smoking** | | | | | | |
| Yes | 7 | 1.13 | 1.07–1.21 | <0.001 | 7.7 | 0.369 |
| No | 4 | 1.47 | 1.07–2.01 | 0.016 | 90.3 | 0.000 |
| **Prostate cancer** | | | | | | |
| **Study population** | | | | | | |
| North America | 3 | 1.03 | 0.89–1.18 | 0.728 | 36.9 | 0.205 |
| Asia | 1 | 1.14 | 0.99–1.31 | 0.065 | – | – |
| Europe | 3 | 1.73 | 1.39–2.16 | <0.001 | 39.8 | 0.190 |
| **Study design** | | | | | | |
| Prospective study | 7 | 1.26 | 1.03–1.54 | | 72.9 | 0.001 |
| **Dental status** | | | | | | |
| Self-reported | 3 | 1.12 | 0.86–1.47 | 0.395 | 70.6 | 0.034 |
| Dental examination | 4 | 1.44 | 1.02–2.02 | 0.036 | 77.1 | 0.004 |
| **Follow-up (years)** | | | | | | |
| ≥10 | 6 | 1.16 | 0.97–1.38 | 0.100 | 60.8 | 0.026 |
| <10 | 1 | 1.84 | 1.36–2.49 | <0.001 | – | – |
| **Adjust for smoking** | | | | | | |
| Yes | 5 | 1.11 | 1.01–1.22 | 0.035 | 43.7 | 0.130 |
| No | 2 | 1.95 | 1.46–2.61 | <0.001 | 43.8 | 0.182 |
| **Lung & bronchus cancer** | | | | | | |
| **Study population** | | | | | | |
| North America | 8 | 1.34 | 1.24–1.46 | <0.001 | 46.9 | 0.068 |
| Asia | 3 | 1.18 | 0.72–1.93 | 0.502 | 59.8 | 0.083 |
| Europe | 3 | 1.11 | 0.64–1.95 | 0.709 | 77.6 | 0.004 |

**Table 2** (*continued*)

| Subgroups | No. of studies | Meta-analysis | | | Heterogeneity | |
|---|---|---|---|---|---|---|
| | | HR | 95%CI | *P*-value | I²(%) | *P*-value |
| **Study design** | | | | | | |
| Prospective study | 11 | 1.30 | 1.10-1.53 | 0.002 | 68.5 | <0.001 |
| Retrospective study | 3 | 1.36 | 1.14-1.63 | 0.001 | 34.4 | 0.218 |
| **Dental status** | | | | | | |
| Self-reported | 6 | 1.31 | 1.21-1.42 | <0.001 | 0.0 | 0.635 |
| Dental examination | 8 | 1.33 | 0.94-1.88 | 0 | 75.7 | <0.001 |
| **Follow-up (years)** | | | | | | |
| ≥10 | 9 | 1.28 | 1.18-1.38 | <0.001 | 46.5 | 0.060 |
| <10 | 5 | 1.20 | 0.81-1.77 | 0.368 | 78.8 | <0.001 |
| **Adjust for smoking** | | | | | | |
| Yes | 10 | 1.35 | 1.25-1.46 | <0.001 | 44.6 | 0.062 |
| No | 4 | 1.10 | 0.76-1.59 | 0.618 | 69.3 | 0.011 |
| **Colorectal Cancer** | | | | | | |
| **Study population** | | | | | | |
| North America | 7 | 1.08 | 0.98-1.19 | 0.134 | 18.2 | 0.291 |
| Asia | 4 | 1.22 | 0.97-1.55 | 0.089 | 91.4 | <0.001 |
| Europe | 2 | 1.10 | 0.68-1.78 | 0.686 | 64.9 | 0.058 |
| **Study design** | | | | | | |
| Prospective study | 11 | 1.19 | 0.99-1.43 | 0.063 | 81.8 | <0.001 |
| Retrospective study | 2 | 1.24 | 1.16-1.33 | <0.001 | 0.0 | 0.327 |
| **Dental status** | | | | | | |
| Self-reported | 8 | 1.19 | 1.02-1.39 | 0.030 | 87.2 | <0.001 |
| Dental examination | 5 | 1.17 | 0.92-1.48 | 0.203 | 26.4 | 0.236 |
| **Follow-up (years)** | | | | | | |
| ≥10 | 9 | 1.21 | 0.99-1.48 | 0.065 | 85.4 | <0.001 |
| <10 | 4 | 1.24 | 1.16-1.33 | <0.001 | 35.9 | 0.182 |
| **Adjust for smoking** | | | | | | |
| Yes | 10 | 1.20 | 1.13-1.26 | <0.001 | 45.2 | 0.058 |
| No | 3 | 1.08 | 0.71-1.63 | 0.718 | 90.3 | <0.001 |
| **Total cancer** | | | | | | |
| **Study population** | | | | | | |
| North America | 6 | 1.13 | 1.09-1.18 | <0.001 | 0.0 | 0.965 |
| Asia | 3 | 1.09 | 0.94-1.27 | 0.243 | 92.6 | <0.001 |
| Europe | 3 | 1.18 | 1.09-1.28 | <0.001 | 47.1 | 0.129 |
| **Study design** | | | | | | |
| Prospective study | 11 | 1.15 | 1.09-1.20 | <0.001 | 67.1 | <0.001 |
| Retrospective study | 1 | 0.84 | 0.58-1.19 | 0.324 | – | – |
| **Dental status** | | | | | | |
| Self-reported | 5 | 1.17 | 1.12-1.22 | <0.001 | 57.1 | 0.054 |
| Dental examination | 7 | 1.08 | 1.03-1.13 | 0.001 | 46.1 | 0.072 |
| **Follow-up (years)** | | | | | | |
| ≥10 | 8 | 1.11 | 1.08-1.15 | <0.001 | 42.1 | 0.097 |
| <10 | 4 | 1.22 | 1.18-1.25 | <0.001 | 49.0 | 0.097 |

| Subgroups | No. of studies | Meta-analysis | | | Heterogeneity | |
|---|---|---|---|---|---|---|
| | | HR | 95%CI | *P*-value | $I^2$(%) | *P*-value |
| **Adjust for smoking** | | | | | | |
| Yes | 8 | 1.13 | 1.09-1.17 | <0.001 | 0.0 | 0.803 |
| No | 4 | 1.18 | 1.06-1.32 | 0.002 | 86.2 | <0.001 |

**Notes.**

CI, confidence interval; HR, hazard ratio.

$P = 0.000$). Similar to prostrate cancer, we found that periodontal disease increased the risk of lung & bronchus cancer incidence under certain conditions in the North American population, self-reported diagnosis of periodontal disease , follow-up period of more than 10 years, and adjustments for smoking, but the association was not affected by study design type.

Among the 13 studies on periodontal disease and colorectal cancer incidence, seven studies were from North America and showed an HR of 1.08 ($I^2 = 18.2$, 95% CI [0.98–1.19], $P = 0.134$), and five studies in which periodontal disease was diagnosed by dental examination showed an HR of 1.17 ($I^2 = 26.4$, 95% CI [0.92–1.48], $P = 0.203$). We found that periodontal disease increased the risk of colorectal cancer incidence under certain conditions in retrospective study, self-reported diagnosis of periodontal disease , follow-up period of less than 10 years, adjustments for smoking, but the association was not affected by study design type.

Among the 12 studies on periodontal disease and total cancers incidence, three studies were from Asia and showed an HR of 1.09 ($I^2 = 92.6$, 95% CI [0.94–1.27], $P = 0.243$), and eight studies with adjustments for smoking showed an HR of 1.13 ($I^2 = 0$, 95%CI [1.09–1.17], $P = 0.001$). We concluded that periodontal disease increased the risk of total cancer incidence under certain conditions in the North American and European population, prospective study design, but the association was not affected by the diagnostic method for periodontal disease, follow-up period, adjustment for smoking.

## Association between periodontitis and cancer mortality

We also analyzed the relationship between periodontal disease and cancer mortality (Fig. 4). A relationship between periodontal disease and total cancer mortality was observed in four studies, and the HR ranged from 1.24 to 1.58; the pooled HR was 1.40, with no heterogeneity ($I^2 = 0$, $P_Q = 0.718$), indicating that the result is reliable. Although the results of current meta-analysis suggest that periodontal disease may increase the risk of breast (HR = 1.25, 95%CI [0.83–1.88], $I^2 = 0.0\%$, $P_Q = 0.809$), prostate (HR = 1.78, 95%CI [0.92–3.46], $I^2 = 0.0\%$, $P_Q = 0.962$), lung & bronchus(HR = 1.48, 95%CI [0.93–2.35], $I^2 = 50.8\%$, $P_Q = 0.154$), colon & rectum (HR = 1.66, 95%CI [0.44–6.27], $I^2 = 76.7\%$, $P_Q = 0.083$) cancer mortality, the overall estimate was not significant because of high levels of uncertainty or moderate heterogeneity. However, we did not perform a subgroup analysis because there were only two studies.

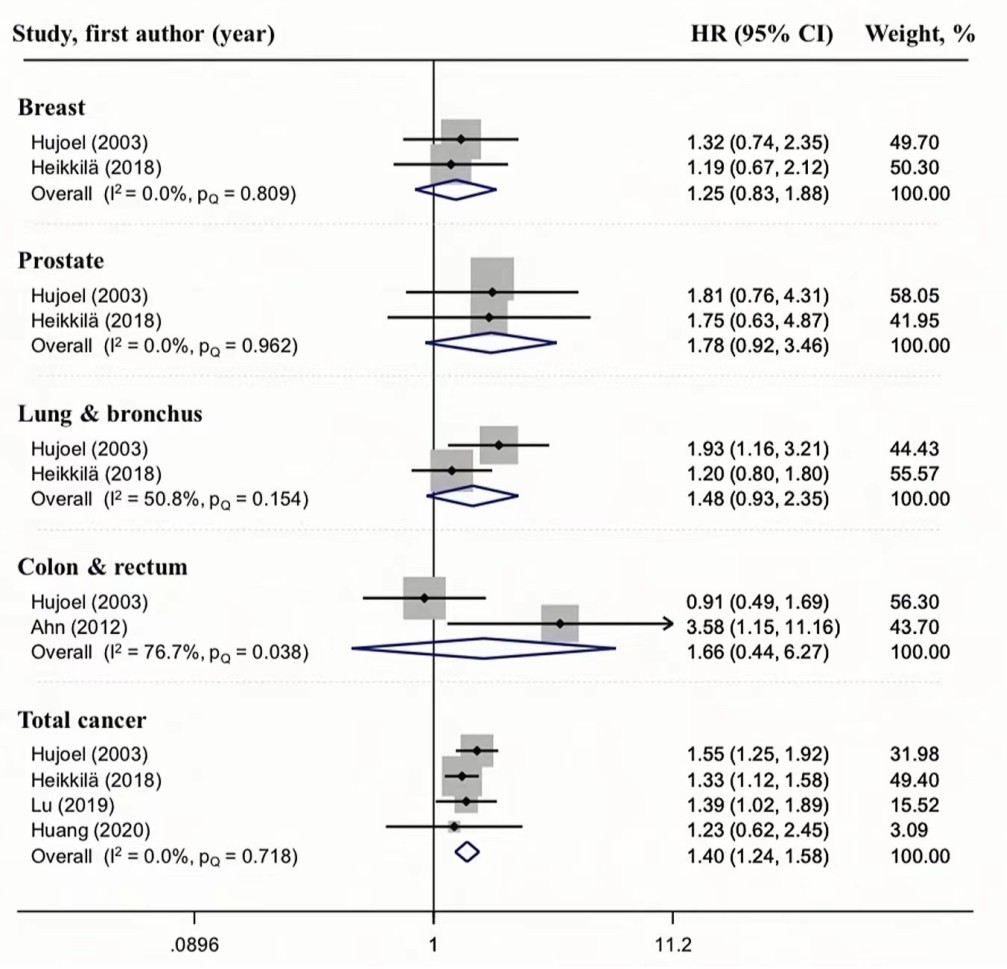

**Figure 4** Forest plots of the association between periodontitis and cancer mortality.

## Publication bias in meta-analysis

Finally, we performed Begg's and Egger's tests to evaluate publication bias. The results of the two tests are shown in Table 3. We conducted Begg's test in the case of missing items; the *P* values of the test results after a continuity correction based on Kendall's score in the remaining studies ranged from 0.133−1.000. All the *P* values were greater than 0.05, indicating that there was no publication bias. We also conducted Egger's test in the case of missing items, the *P* values of the results of assessing whether the intercept of the remaining items was ranged from 0.147−0.860. All the *P* values were greater than 0.05, indicating no publication bias. The results of Begg's and Egger's tests were consistent, indicating that there is no publication bias in this meta-analysis.

## DISCUSSION

So far, there are a few relevant meta-analyses assessing the association between periodontal disease and breast (*Shi et al., 2018*), prostate (*Wei et al., 2021*), lung (*Wang et al., 2020*),

**Table 3  Publication bias in meta-analysis.**

| Publicaton bias (*P*-value) | Cancer incidence | | Cancer mortality | |
|---|---|---|---|---|
| | Begg's test | Egger's test | Begg's test | Egger's test |
| Breast cancer | 0.640 | 0.328 | 1.00 | – |
| Prostate cancer | 0.133 | 0.147 | 1.00 | – |
| Lung & bronchus cancer | 0.767 | 0.574 | 1.00 | – |
| Colorectal cancer | 0.584 | 0.357 | 1.00 | – |
| Total cancer | 0.855 | 0.387 | 1.00 | 0.860 |

and colorectal (*Li et al., 2021*) cancers. Additionally, there are meta-analyses that have revealed that infections caused by periodontal pathogens increase the risk of total cancers incidence (*Michaud et al., 2017*) and mortality (*Romandini et al., 2021a*). However, no one study has described the correlation between periodontal disease and multiple types of cancer in terms of incidence and mortality comprehensively and in detail. Although our results are mostly consistent with those of the previous meta-analyses, our analysis is the most comprehensive because it includes new studies and the latest results, and we reviewed nearly 20 years of reliable research. The sensitivity analysis further proved the accuracy of our results. The subgroup analysis helped us in better analyze the sources of heterogeneity.

On reviewing the 27 studies included in this meta-analysis, we preliminarily found that compared with healthy individuals, patients with periodontal disease were 1.14 times more susceptible to breast cancer, 1.26 times more susceptible to prostate cancers, 1.3 times more susceptible to lung and bronchial cancers, and 1.18 times more susceptible to colorectal cancer. Through a meta-analysis of eight studies, *Shi et al. (2018)* found that periodontal disease increased the susceptibility to breast cancer (risk ratio [RR] = 1.19, 95% CI [1.11–1.26], $I^2 = 17.6\%$, $P = 0.30$). This result is consistent with ours. *Wei et al. (2021)* analyzed seven studies, and the results showed that periodontal disease was significantly associated with prostate cancer (RR = 1.17, 95% CI [1.07–1.27], $I^2 = 5.8\%$, $P = 0.001$); however, the results of the subgroup analysis showed that some of the 95% CIs fell on both sides. After 18 years of follow-up, *Hiraki et al. (2008)* concluded that periodontal disease did not increase the risk of developing colorectal cancer (HR = 1.06, 95% CI [0.91–1.24]). Based on a analysis of three cohort studies, *Ren et al. (2016)* also came to a conclusion that there is no relationship between oral health and colorectal cancer. By contrast, one study reported that women with moderate or severe periodontal disease were at a higher risk of colorectal cancer; there were no adjustments for smoking status, body mass index, or alcohol consumption (*Momen-Heravi et al., 2017*). Another study (*Lee et al., 2018*) found that periodontitis increased the risk of colorectal cancer among male patients, former or current smokers, and patients with an alcohol intake above the moderate level. Similarly, the association between periodontal disease and lung cancer is very controversial. *Wang et al. (2020)* analyzed six high-quality studies and found that periodontal disease greatly increased the risk of lung cancer (HR = 1.40, 95% CI [1.25–1.58], $I^2 = 8.7\%$). Two studies (*Michaud et al., 2008*; *Hiraki et al., 2008*) also obtained the same result after adjustments for smoking-related variables, but they obtained the opposite result when they analyzed

patients who never smoked cigarettes, but were pipe or cigar smokers. *Xiao et al. (2020)* found that periodontal bacterial infection increased the incidence of total cancer (OR = 1.25, 95% CI [1.03–1.52], $I^2 = 71\%$, $P = 0.02$), but the results of the subgroup analysis showed that some of the 95% CIs fell on both sides.

This meta-analysis also concluded that there is a clear correlation between periodontal disease and total cancer mortality. Patients with periodontal disease have a 1.4 times higher risk of dying from total cancer than those without periodontal disease. *Romandini et al. (2021b)* reviewed 57 studies and concluded that periodontal disease increased the risk of all-cause mortality, including cancer mortality (RR = 1.38, 95% CI [1.24–1.53], $I^2 = 29.8\%$, $P < 0.001$), which is consistent with our result.

We found that the correlation between periodontal disease and breast cancer incidence was maintained after the subgroup analysis. While performing the meta-analysis, we found greater heterogeneity in the included studies; subsequently, we performed a subgroup analysis of factors, such as study population, study design, dental status, follow-up period, and adjustments for smoking, which might affect heterogeneity. First, we found that, compared with studies from countries in Europe and Asia, those from North America were less heterogeneous. Second, prospective studies were more heterogeneous. In fact, the two study designs have their own advantages and disadvantages. It is recommended to adopt a prospective follow-up study design if the purpose of a study is to evaluate the effect of a certain treatment method. It is recommended to choose a retrospective follow-up study design if the purpose of a study is to explore and explain a certain phenomenon (*Vandenbroucke, 2008*). The advantage of the prospective design is the accuracy of data including exposures, confounding factors, and endpoints, but these studies require a lot of time and financial resources. The merit of retrospective studies is the time saved by the direct use of existing data, but these studies are susceptible to selection bias. The limitation of retrospective studies is that it is not possible to establish causal effects. The results may be influenced by confounding factors expect to the exposure factors, which are often not accurately measured or may even be unknown (*Euser et al., 2009*). Third, the heterogeneity of the studies, which was observed over a period of 10 years, was relatively low, indicating that the results obtained from the studies with long follow-up periods are relatively accurate and consistent. In the subgroup analysis, six studies with follow-up periods of over 10 years demonstrated a 1.11-fold increase in breast cancer risk, with low heterogeneity ($I^2 = 14.4\%$). Among the eight studies included in Tingting's meta-analysis (*Shi et al., 2018*), five studies had a follow-up period of more than 10 years and revealed a 1.38-fold increase in breast cancer risk through a subgroup analysis with minimal heterogeneity ($I^2 = 7.0\%$). Fourth, among the studies we included, the diagnosis of periodontal disease was self-reported in nine. In many of the previous studies, the diagnosis of periodontal disease was based on self-reported patient data or administrative data. Although this approach may sound unreliable, it can save time. In addition, the manpower and financial resources required for a periodontal examination can be saved. Moreover, self-reported diagnosis of periodontal disease has been proven to be acceptable, with a diagnostic OR of 1.4 (95% CI [0.9–2.2]) for the question on bleeding gums and

of 11.7 (95% CI [4.1–33.4]) for the question on tooth mobility, it was also found to be suitable in large-scale epidemiological studies (*Abbood et al., 2016*).

Smoking is a risk factor of many diseases. A previous study has confirmed that smoking is an independent risk factor for periodontal and respiratory diseases, including lung cancer(OR=2.12, 95% CI [1.32–3.42]) (*Waziry et al., 2017*). Therefore, smoking should be excluded as an confounding factor in an analysis of the association between periodontal disease and lung cancer. Of the 27 studies included in this meta-analysis, adjustments for smoking were performed in 18 studies, and 10 of the 14 studies on periodontitis and lung cancer involved adjustments for smoking. The probability of the development of lung and bronchial cancer in patients with periodontal disease was 1.35 times higher than that in individuals with healthy periodontal tissues, but the statistical heterogeneity was great. In the review by *Michaud et al. (2017)*, based on five studies after adjustments for smoking, positive association was found between periodontal disease and lung cancer, with a pooled RR of 1.33 (95% CI [1.19–1.49], $I^2 = 0\%$, $P = 0.58$), this is consistent with our conclusion, and the values are close to ours. In a prospective study by *Tai et al. (2019)*, smoking was not excluded as a disincentive, therefore, the polled HR (HR = 1.9, 95% CI [1.08–3.35], $P = 0.0267$) was significantly higher than ours.

There are several limitations in this meta-analysis. First, although most confounding factors were adjusted for in the included studies, few studies adjusted for other oral health problems and stress, which may affect the results of these studies. Second, genetic factors play a very important role in the occurrence of cancer. While diagnosing the patients' condition, few researchers noted whether they had family history of cancer, therefore, this important influencing factor was most likely ignored in the data analysis, which would also affect the final results.

## CONCLUSIONS

This meta-analysis revealed that periodontal disease is significantly associated with the risk of the incidence of breast cancer and that periodontal disease may increase the risk of total cancer mortality. Our results prompt the general public to pay attention to periodontal health and to receive periodontal treatment in a regular and timely manner. In conclusion, our results suggest that patients with periodontal disease should pay special attention to breast cancer screening during regular general health examinations. Apart from breast ultrasound, a mammography examination conducted once every 1–2 years would be helpful for the early detection and early treatment of breast cancer. Of course, attention should also be paid to other related cancers, such as lung, colorectal, and prostate cancers. However, well-designed studies, populations from diverse geographic regions, and adjustments for exact confounding factors are needed to confirm our findings and to further explore the relationship between periodontal disease and cancers incidence and mortality.

### Funding

The authors received no funding for this work.

### Competing Interests

The authors declare there are no competing interests.

### Author Contributions

- Kaili Wang conceived and designed the experiments, performed the experiments, analyzed the data, prepared figures and/or tables, authored or reviewed drafts of the article, and approved the final draft.
- Zheng Zhang conceived and designed the experiments, performed the experiments, analyzed the data, prepared figures and/or tables, authored or reviewed drafts of the article, and approved the final draft.
- Zuomin Wang conceived and designed the experiments, authored or reviewed drafts of the article, and approved the final draft.

### Data Availability

The raw measurements are available in the Supplementary Files.

### Supplemental Information

Supplemental information for this article can be found online at http://dx.doi.org/10.7717/peerj.14320#supplemental-information.

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
