# Peer review of "Assessment of the association between periodontal disease and total cancer incidence and mortality: a meta-analysis"

_PeerJ, doi:10.7717/peerj.14320_

## Round 0.1 · original submission · Major Revisions

Dear authors,
Thank you for submitting your manuscript to PeerJ. Please follow up with all corrections as suggested by the reviewers. Your immediate action will help us to speed up the decision. Thank you

Reviewer 1 ·

Basic reporting

Please do check the below statement in the background column of the abstract.
“However, there is still no specialized meta-analysis that deeply assesses
the association between periodontal disease and cancer incidence and mortality. Thus, we conducted this meta-analysis.” May I Suggest this statement be paraphrased as “…. that assesses the association between periodontal disease and cancer incidence and mortality in-depth”.

“Results: Twenty-seven studies
were included in this meta-analysis. No article presented poor quality.” Please change the framing of this particular sentence.
The text is interspersed with minor grammatical errors
I would recommend that grammer and language be improved. Examples include lines 25,33 ,39 ,41.91.108,109,207 ,210,252.253 254,.255.256,257,258,275,278,291,295, 296

Please alter the phrasing of ‘smoking being an interference factor”. May I suggest that “confounding factor” is a better option( 278).

Experimental design

Aims and objectives can be more clearly elucidated. The article can clearly state if it is looking for an association between periodontal disease and cancers in general or any particular cancer.

Patients with poor access to health care facilities due to economic conditions or other factors may also have periodontal disease. They may also present in more advanced stages of cancer to healthcare workers. The authors can elaborate on these factors?
Were oral cancers included in the study?

The authors can analyze and elaborate on the age component of the included studies. With increasing age, the incidence of cancer mortality of cancer increases. Elderly people are known to have periodontal disease and also have a higher incidence of cancer in general.

Validity of the findings

The periodontal disease seems to be associated with an increased incidence of cancer, however, the strength of such association can be further clarified.

There seems to be a lot of emphasis on breast cancer associated with periodontal disease when the meta-analysis found a stronger correlation between prostate cancer and bronchial and lung cancer

The authors can describe the implications of the study findings for health care services. Do patients with the periodontal disease require more cancer screening services? The analysis may probably answer this question

Additional comments

The overall quality of the review and meta-analysis is good

Reviewer 2 ·

Basic reporting

In general, the manuscript was ambiguous. Room for improvement.

Literature review: Scientific rigour was unclear. Gaps to justify for the research paper. Is the relationship between periodontitis and these cancers well established? Only quoting 2 studies on association between periodontitis and oral cancer. None other references for other type of cancers. Suggestion: To expand on literature to justify the study.

Professional article structure - not following standard sequence of reporting, formatting can be improved e.g. use of comma, full stop etc, use scientific terms e.g. gingiva rather than gums. Suggestion: To send for proof reading.

Experimental design

1. Methodology section was unclear.
- Not following a standard sequence of reporting
e.g. Inclusion and exclusion criteria should come before Search Strategy / Literature Search.

2. Literature search
- Any hand search to complement the process
- Explain how the broad terms of cancer, tumour and carcinoma can be narrowed down to the intended 4 types of cancer in interest.

3. Criteria for inclusion:
- State the publication language used.
- Justify why case control studies were not included
- Specify (i) the disease that poses the exposure and (ii) disease that poses as the outcome of interest.
- Explain what action taken to multiple study reports with the same study population. Which one will be taken into consideration?

4. Data extraction
- Need to describe the study selection process in detailed.
- Was it based on title / abstract or full paper?
- Was it carried out independently by the 2 authors or how?
- Any removal of duplicates?

5. State case definition for periodontitis used. The recently classification was introduced in 2017. This must have implications on the data collected from 2003 to 2020.

6. For Newcastle Ottawa scale
- State the domains evaluated by this tool.
- State the inter-examiner agreement score.

Validity of the findings

Statistical analysis
1. RR and OR test were mentioned in the inclusion criteria. Justify why no analysis on RR and OR reported.
2. Describe Begg's and Egger's tests in the methodology. Usually, publication bias among included studies is visually examined using funnel plots.

Results:
1. To state reasons for exclusion in the text.
2. Describe the indicators for poor and good quality studies
3. Discuss the key findings based on the objectives
4. Justify for meta-analysis without systematic review first.
5. Publication bias: Not much explanation was given before deducing the conclusion. Readers need to understand the domains and process involved, and how the conclusion was drawn.

Discussion:
Some repetition of literature review in the discussion. Please remove.

Additional comments

I believe this is an original study conducted by the experts. Well done!

However, there are few areas in 2 and 3 sections (see above comments) that must be clarified to ensure that proper steps have been taken to conduct the meta analysis and no flaws were involved. All concerns in methodology and results sections must be addressed. This is especially when the conclusions made can be alarming and raised concerns by the public.

Need to proof reader by an English speaking background person to improve the manuscript.
Authors need to pay attention to professional article structure format.

Thank you.

---

## Round 0.2 · accepted · Accept

I am happy with all the corrections although the second reviewer did not respond to the second review invitation. All the comments by reviewers were adequately addressed. The manuscript is ready for publication.

Reviewer 1 ·

Basic reporting

No further comments

Experimental design

No further comments

Validity of the findings

No further comments